

# Face-voice based multimodal biometric authentication system via FaceNet and GMM

Bayan Alharbi and Hanan S. Alshanbari

Department of Computer Science, Umm Al-Qura University, Makkah, Saudi Arabia

## ABSTRACT

Information security has become an inseparable aspect of the field of information technology as a result of advancements in the industry. Authentication is crucial when it comes to dealing with security. A user must be identified using biometrics based on certain physiological and behavioral markers. To validate or establish the identification of an individual requesting their services, a variety of systems require trustworthy personal recognition schemes. The goal of such systems is to ensure that the offered services are only accessible by authorized users and not by others. This case study provides enhanced accuracy for multimodal biometric authentication based on voice and face hence, reducing the equal error rate. The proposed scheme utilizes the Gaussian mixture model for voice recognition, FaceNet model for face recognition and score level fusion to determine the identity of the user. The results reveal that the proposed scheme has the lowest equal error rate in comparison to the previous work.

# INTRODUCTION

Information security is involve the protection of information's integrity, availability and confidentiality in all forms. There are numerous technologies and approaches that can aid with information security management. Biometric systems, however, have developed to help with some areas of data security. Biometric authentication aids in identification, authentication, and non-repudiation in the area of information security.

As a method of providing personal identification, biometric authentication has gained popularity. In many applications, a person's identification is critical, and this is a major concern in society, as evidenced by the surge in identity theft and credit card fraud in recent years. In today's highly linked world, the use of pin identification, token-based systems and individual passwords is constrained by inherent faults. When opposed to a template, biometrics are utilized to determine the identity of a user from a captured sample, which is useful when trying to identify certain people based on certain features. A user can be authenticated and verified using one or more of three basic approaches: knowledge-based security involves using a code or password, and possession-based security involves using a single unique "token," like a security tag or a card. For sufficient validation, standard validation systems frequently use numerous sample inputs, such as specific sample features. This aims to improve security by requiring many diverse samples, such as security tags

Corresponding author
Hanan S. Alshanbari,
hsshanbari@uqu.edu.sa

and codes, as well as sample dimensions. As a result, biometric authentication claims to be able to create a solid one-to-one connection between a person and a piece of information. In this paper, we provide an enhanced scheme of face-voice based multimodal biometric authentication to reduce the Equal Error Rate (EER).

## Types of biometrics

The biometric system can be divided into two categories:

1. Unimodal biometric system: A single biometric feature (either physical or behavioral trait) is used to identify the person in a unimodal biometric system. Biometric system based on face, palmprint, voice, or gait. Figure 1 is an example of a Unimodal biometric system based on Iris authentication.
2. Multimodal biometric system: It is a biometric system that combines data from numerous sources. For instance, a biometric system based on a person's face and gait, or a person's face and speech, etc.

One human body attribute is measured and evaluated by unimodal biometric systems. Unimodal biometrics have a number of drawbacks, including:

- **Noise in sensed data:** A biometric system's identification rate is highly dependent on the quality of the biometric sample.
- **Non-universality:** A biometric modality is said to be universal if any individual in a population can offer it for a particular system. Not all biometric modalities, however, are genuinely global.
- **Absence of individuality:** Individuals' biometric data may reveal similarities between their attributes (*Ammour, Bouden & Boubchir, 2018*).
- **Intra-class variation:** Biometric data collected during an individual's training procedure for the purpose of creating a template will differ from biometric data collected during the test process for the same user. These differences could be attributable to the user's inadequate engagement with the sensor (*Kabir, Ahmad & Swamy, 2018*).
- **Spoofing:** While it may appear difficult to steal a person's biometric modalities, spoofed biometric modalities can always be used to overcome a biometric system.

To solve these drawbacks, one alternative is to combine many biometric modalities into a single system, which is known as a multi-biometric system (*Kabir, Ahmad & Swamy, 2018*; *Matin et al., 2017*).

## Multimodal biometric system

Multimodal biometric systems are biometric systems that provide person identification based on information acquired from numerous biometric features. Figure 2 displays a block diagram of a multimodal biometric system.

The multimodal biometric system has several benefits over a unimodal biometric system, which are described below:

1. A multimodal biometric system aquires more than one form of information as compared to a unimodal biometric system, resulting in a significant improvement in matching accuracy.

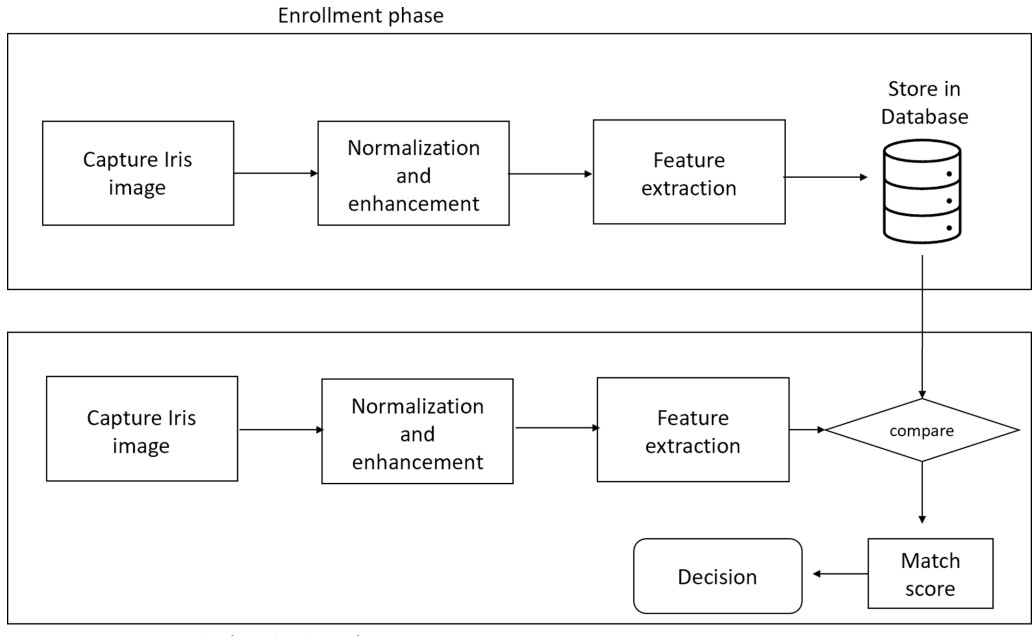

**Figure 1** Block diagram of unimodal biometric system based on Iris authentication.

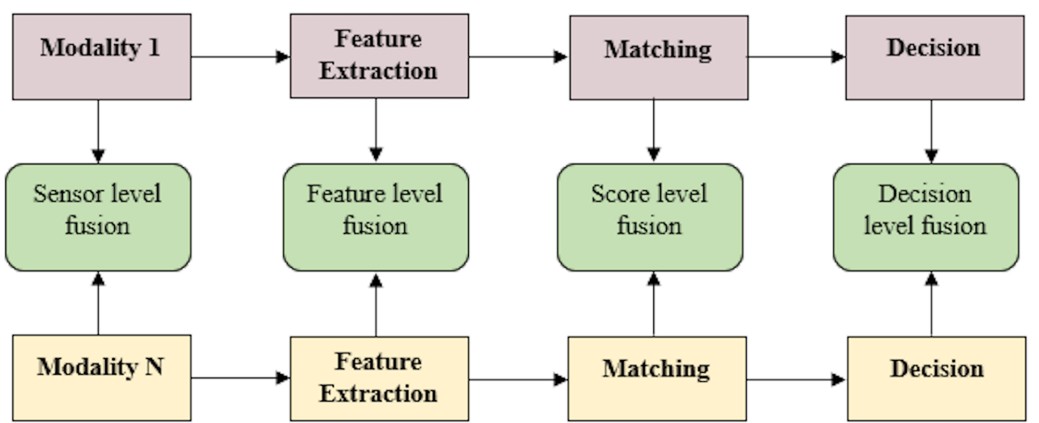

**Figure 2** Block diagram of multimodal biometric system.

2. Multimodal biometric systems can handle the issue of nonuniversality (*e.g.*, with a unimodal biometric system, 2% of the population does not have a good fingerprint *Ross, Nandakumar & Jain, 2006*) by accommodating a large number of users. Even if a user lacks a single valid biometric characteristic, they can nonetheless be enrolled into a system using another valid biometric trait. A higher level of flexibility can be accomplished by registering the user and acquiring his many attributes, with only a subset of the gained features being verified.

3. Importer attacks are less likely with multimodal biometric systems. Spoofing a legitimate person enrolled in a multimodal biometric system is extremely complicated.

4. Multimodal biometric systems are immune to noise in sensed data, which means that if information from a single biometric trait is contaminated by noise, we can do verification using another biometric trait from the same subject.

5. In situations where a single biometric feature is insufficient, these technologies can assist in continuous monitoring or tracking of the individual. For example, tracking a person's face and walk at the same time. The rest of the paper is organized as follows: Section two previous related studies are summarized. The utilized scheme and methodology is described in section three. In section four the experimental results with discussion are presented. Finally, section five summarizes and clarifies the research contributions, implications, and suggested future works.

## RELATED WORK

Previous research and recently published works are presented in this section. In effort to mitigate the (hackable password, weak password, difficulty of remembering complex passwords) several researchers attempted to introduce a unique biometric recognition technology based on human characteristics (face, voice, iris, fingerprint). A plethora of approaches are presented for biometric authentication. The earliest efforts utilized a single human body feature to measure and evaluate by unimodal biometric systems. These papers (*Lin & Kumar, 2018*; *Huvanandana, Kim & Hwang, 2000*; *Uchida, 2005*; *Batool & Tariq, 2011*) identify users solely based on finger biometrics with ranging accuracy rates. With the advancements of the technology the multimodal biometric identification are introduced, to evade vulnerabilities of malicious acts and to add a layer of complexity to hacking attempts and lessen the unimodal drawbacks.

Several multimodal combination are introduced. *Brunelli & Falavigna (1995)* proposed a biometrics based on combined facial and fingerprint biometrics. Their work was extended by *Kittler & Messer (2002)* to include face, finger print and hand geometry biometrics. *Frischholz & Dieckmann (2000)* proposed BioID system which identifies users by face, voice, and lip movement. The multimodal system identifies users based on the biometric characteristics, in addition it has a verification mode which a user enters their name or number then the system verifies the user by their biometric characteristic. *Joseph et al. (2021)* improved multimodal biometric authentication in cloud computing environment by combining the characteristics of fingerprints, palm and iris prints and generating a unique secret key then, converting it to a hashed string. To evade against hill climbing attacks (*Sarier, 2021*) proposed a new multimodal biometric authentication (MBA) for mobile edge computing (MEC) protocol (MBA-MEC). The proposed system functions on two traits fingerprints and face. Despite, improvement of the security phone devices may experience delay due to the device restricted computation capacity.

Another combination of biometrics are introduced based on face and voice. The introduction of face and voice biometrics is made possible by the fact that they are simple to collect quickly and accurately using inexpensive technologies. A face recognition

model based on fusion technique and transform domain provided by *Halvi et al. (2017)* contained two transform domain techniques, Fast Fourier Transform (FFT) techniques and the discrete wavelet transformation (DWT) (*Alan, Ronald & Buck, 1989*; *Bracewell, 2000*). Euclidian distance (ED) is used to compare the recovered features from the DWT and FFT in order to compute the parameter performance. The wireless body area network's security is boosted by biometric security (WBAN). WBAN is a subclass of wireless sensor networks (WSNs) and is a wireless network created for medical applications (*Dodangeh & Jahangir, 2018*; *El-Bendary, 2015*; *Qi, Chen & Chen, 2018*).

*Poh & Korczak (2001)* developed a hybrid prototype for person authentication that included facial and text-dependent voice biometrics. In this prototype, the features vector used moments to extract face information and wavelets to extract speech information. Two distinct multi-layers are used to classify the resulting characteristics. The findings of this system have an equal error rate (EER) of 0.15 percent for face recognition and 0.07 percent for speech recognition. In *Elmir, Elberrichi & Adjoudj (2014)* work, a hierarchical multimodal technique based on face and speech is proposed for user authentication. Mel frequency cepstral coefficients (MFCCs) features that are taken from the speech and Gabor filter bank are used to create the face features vector. The similarity of the planning coefficients is compared using the Cosine Mahalanobis distance (CMD). The results had an EER of 1.02 percent for face differentiation, 22.37 percent for voice differentiation, and 0.39 percent for multimodal differentiation. Another effort is proposed by *Zhang et al. (2020)* utilizing voice and face biometrics to develop an Android-based multimodal biometric authentication system. The main contribution of this work is reducing space and time complexity.

In *Kasban (2017)* work The features vector for voice is extracted using the MFCCs, linear prediction coefficients (LPC), and linear prediction cepstral coefficients (LPCC) features, whereas the features vector for face differentiation is extracted using Principal Components Analysis (PCA), Latent Dirichlet Allocation (LDA), and the Gabor filter. Log likelihood ratio (LLR) was used to bring all of these aspects together. The results had an EER of 1.95 percent for face differentiation, 2.24 percent for voice differentiation, and 0.64 percent for multimodal differentiation. The MFCC features are used to extract the features vector for speech in this study (*Soltane, 2015*), whereas Eigenfaces are used to extract the features vector for face differentiation. The merging of these aspects was achieved with the help of Gaussian mixture model (GMM). EERs of 0.28125 percent for multimodal differentiation, 0.00539 percent for voice differentiation, and 0.39995 percent for face differentiation were attained as a result of the research. In *Abozaid et al. (2019)* work, according to the results of the speech recognition technique, the best results were obtained by simulating the Cepstral coefficients using a GMM classifier scenario with a 2.98 EER. With an EER of 1.43, it was determined that the PCA with GMM classifier-based face differentiation strategy was the most effective approach for face recognition. The results of the fusion revealed that the lowest EER is produced by the scores fusion (0.62), indicating that it is a promising multimodal fusion strategy. Table 1 outlines different schemes of multimodal biometric using face and voice.

**Table 1 Several multimodal biometric systems utilizing speech and face recognition.**

| Multimodal biometric approach | Extracted features | | Fusion technique | Database | Results (EER %) | | |
|---|---|---|---|---|---|---|---|
| | Face | Voice | | | Face | Voice | Fusion |
| *Poh & Korczak (2001)* | Moments | Wavelet | No Fusion | Persons | 0.15 | 0.07 | – |
| *Chetty & Wagner (2008)* | DCT, GRD, CTR | MFCC | GMM | AVOZES | 3.2 | 4.2 | 0.73 |
| *Palanivel & Yegnanarayana (2008)* | MDLA | WLPCC | GMM | nEWSPAPERS | 2.9 | 9.2 | 0.45 |
| *Raghavendra, Rao & Kumar (2010)* | 2D LDA | LPCC | GMM | VidTIMIT | 2.1 | 2.7 | 1.2 |
| *Elmir, Elberrichi & Adjoudj (2014)* | Gabor filter | MFCC | CMD | VidTIMIT | 1.02 | 22.37 | 0.39 |
| *Soltane (2015)* | Eigenfaces | MFCC | GMM | eNTERFACE | 0.399 | 0.0054 | 0.281 |
| *Kasban (2017)* | PCA, LDA, Gabor filter | MFCCs, LPCs, LPCCs | LLR | PROPOSED | 1.95 | 2.24 | 0.64 |
| *Abozaid et al. (2019)* | Eigenfaces, PCA | MFCC | LLR | PROPOSED | 2.98 | 1.43 | 0.62 |

From the conducted review of the related works we propose a scheme utilzing the GMM model and FaceNet. The FaceNet model has a high accuracy rate therefore was utilized to extract the face embedding in our scheme. The FaceNet model is based on a Siamese neural network this type of network is highly suitable for user authentication applications. It offers more resistant to class imbalance, focuses on learning embeddings (in the deeper layer) that groups together similar classes and notions. As a result, semantic similarity can be learned. From Table 1 we observed the best results for voice recognition are yielded from using a GMM model. In this paper we explore the utilization of GMM and FaceNet for a multimodal biometric authentication system.

## METHODOLOGY

This section discusses the details of construction and architecture of the proposed system for voice and face authentication. The system is divided to three sections (i) voice recognition, (ii) face recognition and (iii) score level fusion. The proposed block diagram for the multimodal biometric fusion method is shown in Fig. 3. The proposed scheme enhances security utilizing collaborative model learning, federated learning (FL) extracts knowledge from decentralized data, improving data security, privacy, and secrecy. Data protection is greatly important these days, therefore decentralizing training is considered promising strategy for maintaining privacy and secrecy (*Lu et al., 2022*). Obtaining voice and facial samples is the initial step. Features are then extracted from the samples and saved in the database. In the process of authenticating the user voice and face samples are also acquired and features are extracted then compared with the database, finally, the score fusion is conducted and the decision is made to whether the user is denied access or accepted. We built a multimodal biometric authentication system utilizing GMM (Gaussian mixture model) and face recognition using FaceNet. In *Abozaid et al. (2019)* the GMM model yielded the lowest EER results for voice recognition. FaceNet model is utilized for face recognition since it has a 93% accuracy (*Schroff, Kalenichenko & Philbin, 2015a*). *Abozaid*

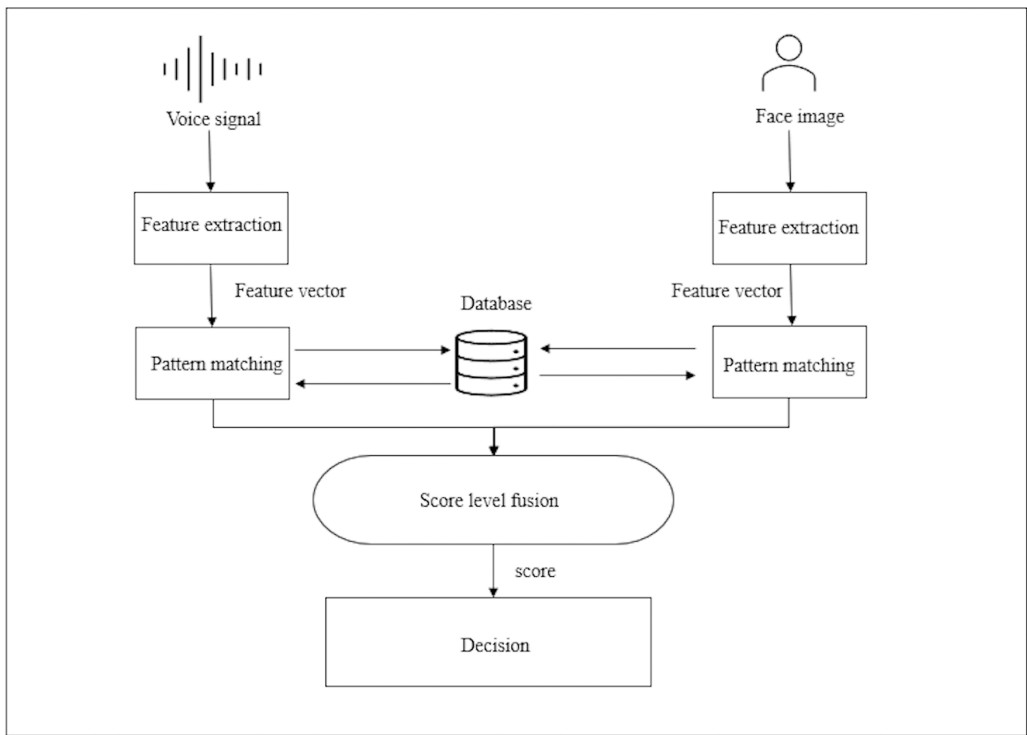

**Figure 3** The proposed multimodal biometric fusion scheme block diagram.

*et al. (2019)* experiments showed that the score level fusion produce high accuracy and low EER results hence, it is utilized in this research.

## Voice recognition

Human voice is unique the arrangement of the teeth, the trachea, the nose, the voice chords, and the way a person amplifies sounds are all factors that contribute to the uniqueness of voice. Such features, when combined, are as unique as fingerprints and cannot be duplicated or transferred (*Khitrov, 2013*). The contact-free application of speech biometrics is what sets it apart from other modalities. Voiceprints, unlike fingerprints, can be taken from a distance. Unlike fingerprinting and vein recognition, there is no requirement to be physically close to the print capture device while using voice. As a result, voice biometrics may be utilized in a far wider range of settings, such as while driving, from another room, or even on mobile devices (*Khitrov, 2013*). Any speech biometric system works on the following principle: the user or caller utters a passphrase, which is collected by the device and compared to a previously saved voiceprint. According to how well the new speech fits the voice-print recorded in the database, a score is generated by the matching technique. For added security, access score thresholds can be pre-set. Match access will be prohibited if a match method generates a low score (*Khitrov, 2013*). Figure 4 illustrates the proposed voice recognition process. Voice recognition process commences by acquiring the voice sample then extracting the features followed by removing noise and finally, verifying the user.

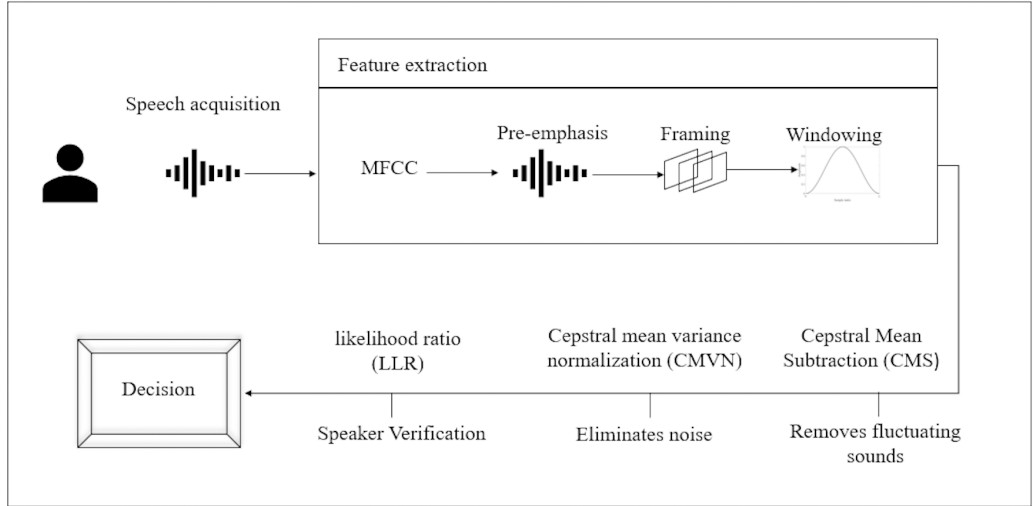

**Figure 4** **Proposed voice recognition process.**

### Gaussian mixture model (GMM)

A parametric probability density function is represented by a Gaussian mixture model, which is the aggregated sum of Gaussian component densities. The probability distribution of continuous measures or features, such as vocal-tract associated spectral data in a speaker recognition system, are typically modeled using GMMs in biometric systems. GMM parameters are estimated from training data using the iterative expectation-maximization (EM) approach or Maximum a Posteriori (MAP) prediction from a learned prior model. The first step in recognition is to scale audio and turn it into MFCC characteristics in order to simplify the model. Then, from the provided feature vector matrix, extract the 40-dimensional MFCC and the delta MFCC features, combine them, and supply the combined data as input to the GMM model.

### Speech acquisition and feature extraction

The features are extracted from the voice signals to be converted to a sequence of acoustic feature vertures which is utilized to identify the speaker. The verification process starts with MFCC parameters.

1.  MFCC parameters
    (a)  Pre-emphasis
        This step enhances high frequencies of the spectrum of the form 1:
        $$x_p(t) = x(t) - ax(t-1) \tag{1}$$
        where $a$ is assigned a value of 0.98.
    (b)  Framing
        Then, the signal is divided into frames. The shift is 10 ms, with a frame length of 20 ms. We utilize framing due to fluctuation in frequencies in a signal over a period of time.

(c) Windowing

Since framing cuts the signals into frames in most cases the start of the following frame usually does not coincide with the finish of the previous frame. Thus, windowing is required to produce an accurate value for the original signal frequency and educe spectral leakage. The Hamming window as follows,

$$w[n] = a_0 - \underbrace{(1-a_0)}_{\simeq a_1} . \cos(\frac{2\pi n}{N}), 0 \leq n \leq N \tag{2}$$

Where $a_0 = 0.53836$.

(d) Fast Fourier Transform (FFT)

The FFT is utilized to convert the signal from its original domain to a representation in the frequency domain. The Cooley–Tukey algorithm is utilized where it divide-and-conquer and recursively breaks down a Discrete Fourrier Transform (DFT) of any composite size into many smaller DFTs of sizes.

(e) Modulus

The magnitude is determined by computing the FFT's absolute value. The power spectrum that has been sampled over 512 points is obtained. Due to the symmetry of the spectrum, only half of those points are relevant.

(f) Mel Filters

There are several of oscillations in the spectrum at that moment, and it is not desired. The size of the spectral vectors must be reduced, hence a smoothing technique is required. As a result, the spectrum is divide by a filterbank, which is a collection of bandpass frequency filters. For frequency localisation, the Bark/Mel scale was used, which is analogous to the human ear's frequency scale. The frequency heard by the human ear is the basis for the measuring unit known as the Mel.

(g) Discrete Cosine Transform (DCT)

The log of the spectrum is transformed using a discrete cosine transform. The MFCCs are obtained, and because the majority of the information is obtained during the first few coefficients, we choose the first few 12. In this step, the MFCC coefficients describing the input signal window is obtained.

(h) Cepstral Mean Subtraction (CMS)

Finally, each vector has the cepstral mean vector subtracted from it. *Furui (1981)*, which is especially important in speaker verification assignments. Cepstral mean subtraction (CMS) is a step that removes slowly fluctuating convolutive sounds.

(i) Cepstral mean variance normalization (CMVN)

By linearly converting the cepstral coefficients to have the same segmental statistics, cepstral mean variance normalization (CMVN) eliminates noise contamination for robust feature extraction. Short utterances, however, are known to decrease the performance of speaker verification tasks.

## Speaker verification

We calculate the claimed identity GMM's score for a sample from the test folder in the enrollment set. By deducting the score from the GMM of the universal background model (UBM), we can obtain the likelihood ratio (*Reynolds, 1997*) for each voice model. We

next compare the score to our threshold of 0.5 and decide whether to accept or reject the speaker's identification.

## Face recognition

Face is an appealing biometric feature since it is simple to gather and has widespread societal acceptance. Facial-recognition technology is highly suited for surveillance applications since it can get a face image from a distance and without the user's consent. The appearance-based facial features, however, that are typically used in the majority of modern face-recognition algorithms have been shown to have weak discrimination abilities and to change over time. Due to hereditary considerations, a tiny percentage of the population can have essentially identical appearances (*e.g.*, identical twins, father and son, etc.), making the face-recognition process more challenging.

The model is based on the FaceNet model that was described by *Schroff, Kalenichenko & Philbin (2015b)* at Google in their 2015 paper. FaceNet is a state-of-the-art neural network for face identification, verification, and clustering. The model The face recognition begins with detecting the face using the Haarcascade classifier, then resizing the region of interest (ROI) and return 128 dimension facial encodings. Figure 5 illustrates the proposed face recognition process. Face recognition process commences by acquiring the face sample by detecting the face then extracting the face embedding followed by predicting the identity and finally, verifying the user.

### Face detection

Face detection is the process of automatically detecting and localizing faces in a photograph by drawing a bounding box around their extent. We utilized the Haar Cascade classifier to perform this task. It is an object detection algorithm (ODA), a method for finding faces in pictures or live video. In their research work, *Viola & Jones (2001)* suggested edge or line detecting features. Finding Haar features, creating integral images, training Adaboost, and classifying the images using a cascading classifier are the four processes of the cascade function for image training (*Priambodo et al., 2021*), as illustrated in Fig. 6. The most crucial step while utilizing the algorithm is that the classification A significant number of datasets with positive facial photos and negative non-face images are required for the training procedure. The Haar Cascade classifier suites the system since only frontal face images are required for detection. For each identified face, the FaceNet model will be utilized to construct a face embedding, and then a linear support vector machine (SVM) classifier model will be developed to predict the identity of that face. After the face is detected the image's region of interest (ROI) is resized to 96*96.

### Face embeddings

The features retrieved from the face are represented by a face embedding, which is a vector. The vectors created for other faces can then be compared. The detected faces from the previous step is loaded and the FaceNet model generates the embeddings for each detected face. To predict an embedding, the image's pixel values must first be appropriately prepared to fulfill the FaceNet model's expectations.

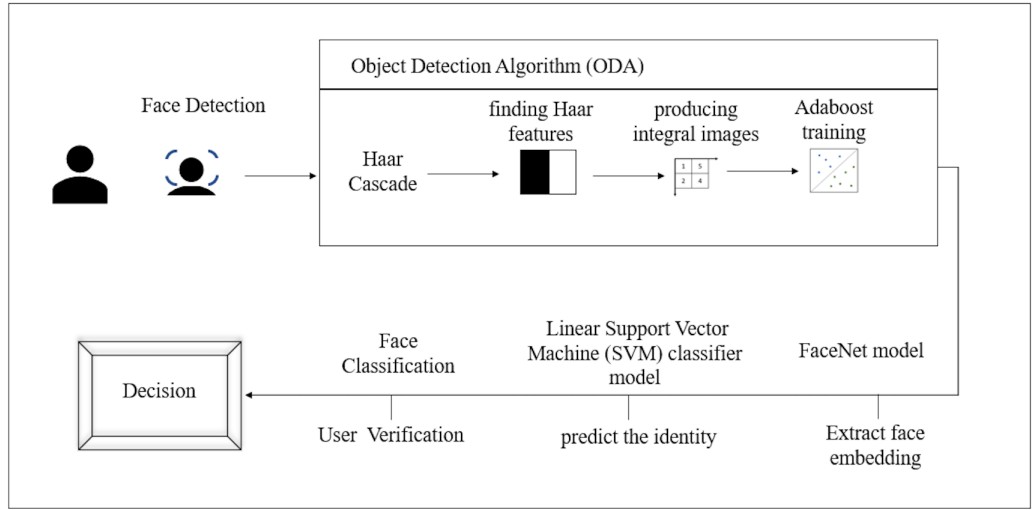

**Figure 5**  **Proposed face recognition process.**

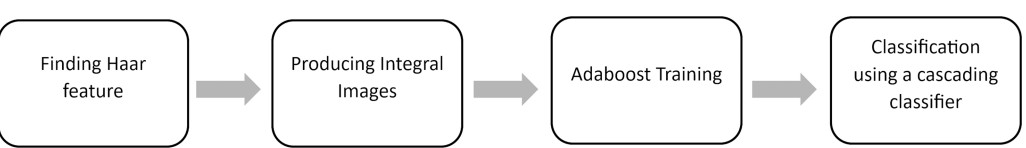

**Figure 6**  **Haar technique.**

### Face classification

In this phase our model classifies face embeddings as one of the users stored in the database. The face embedding vectors are normalized utilizing L2 norm. Since vectors are compared utilizing distance metrics, we normalize the vectors by scaling the values until the length of vector is 1.

## Score level fusion

To determine a person's identity, the match scores produced by various biometric matchers are combined in score-level fusion. Typically, the result of this consolidation technique is the creation of a single scalar score, which is then used by the biometric system (*He et al., 2010*). Since it is easy to acquire and process match scores, fusion at this level is among the most frequently mentioned method in the biometric literature (*Ross & Nandakumar, 2009*). The final step is fusing the recognition scores using score level fusion. The Likelihood Ratio (LLR) computes the total fused score by *Xuan, Xiang & Ma (2016)*, as given in Eq. (3):

$$S = \frac{p(S_{voice}|G).p((S_{face}|G))}{p(S_{voice}|I).p((S_{face}|I))} \tag{3}$$

where $S_{voice}$ is the matching score from the voice recognition technique, $S_{face}$ is the matching score from the face differentiation technique, and $(p(.\text{—}G))$ is the matching

scores probability density function for the actual person, $p(.|I)$ is the matching scores probability density function for the impostor person.

Based on the likelihood of voice identification if the user probability distribution matches the one stored in the database the user proceed with the face identification otherwise access is denied. Next, if both the voice and the face match, and the similarity between the input and recorded encodings of the face recognition database is less than the threshold value (0.5), the user is recognized as permitted.

## Dataset

The AVSpeech dataset (*Ephrat et al., 2018*) is a large-scale audio-visual dataset that contains voice video clips with no background noise. The sections are 3-10 s long, and the soundtrack in each clip belongs to a single speaking individual who is visible in the video. The collection contains over 4700 h of video segments culled from 290,000 YouTube videos, including a wide range of people, languages, and facial expressions.The utilized dataset mimics the real-world conditions since the audio and videos of users are clips from YouTube. The lighting is different for users in several condition white yellow and somewhat dark. As for the audio it is not recorded in a studio environment there is background noise, and the microphones are not high quality like the other datasets. Moreover, this dataset includes different ethnicities and races which make it more suitable to generalize. The dataset is utilized for testing and training. There are seven users with each 100 samples a total of 700 samples. In training one user with an audio sample of 3 s are utilized the rest of the samples are for testing.

## Data analysis

The performance in this research is measured by utilizing the EER to aid in comparing this multimodal biometric scheme with the predecessor schemes. The EER, which is defined as illustrated in Eq. (4), refers to the location at which the FRR and FAR are equal.

$$EER = \frac{FAR + FRR}{2} when \ FAR = FRR \tag{4}$$

where FAR and FRR stand for false acceptance rate and false rejection rate, respectively.

### *Fusion data*

Data fusion requires computing the EER of the fusion of those two classifiers.

- Voice recognition gives as an output a log-likelihood (GMM).
- Face recognition gives a distance in a feature space.

First we convert log likelihood to probabilities by transforming the log-likelihood to a discrete probability with:

$$P(x|\theta_1) = e^{l(x|\theta_1)}. \tag{5}$$

The probability is further normalized with:

$$P(x|\theta_1) = \frac{e^{l(x|\theta_1)}}{\sum_l e^{l(x|\theta_1)}}. \tag{6}$$

The results are too small, so the maximum is factored out by:

$$P(x|\theta_1) = \frac{e^{lmax} \cdot e^{l(x|\theta_1)^{lmax}}}{e^{lmax} \cdot \sum_l e^{l(x|\theta_1)^{lmax}}} = \frac{e^{l(x|\theta_1)^{lmax}}}{\sum_l e^{l(x|\theta_1)^{lmax}}}. \tag{7}$$

Then converting distance to probabilities utilizing the softmax of the negative distance by the formula:

$$P(x_i) = \frac{e^{-d_l}}{\sum_j e^{-d_j}}. \tag{8}$$

Finally, the probabilities are combined with a weighted average. The GMM tends to give extreme probabilities thus, more weight is added on face recognition to counteract this effect.

## RESULTS AND DISCUSSION

In this section we display the obtain results of validating the proposed system. The results are displayed as follows: Fig. 7A is the ROC for the voice recognition which yielded a EER of 0.13. Figure 7B is the ROC for face recognition with 0.22 EER. Lastly, Fig. 7C is the ROC of score level fusion with 0.011 EER. Table 2 compares the results of the proposed scheme and the previous work. Our work significantly reduced the EER for voice, face and fusion.

The GMM model aided in reducing the EER due to, its high accuracy rate. The voice acquisition utilizing the MFCC enables the model to frame and window the voice sample to extract features. CMS is applied to remove the fluctuating sounds and CMVN eliminates noise thus, obtaining higher quality audio sample to identify a user. The final step where the log likelihood is obtained, the speaker is verified based on the threshold of (0.5). As illustrated in Fig. 7B the EER is 0.13 which is the second to lowest obtained value in comparison to the related work as in Table 2.

Face recognition performed highly and the EER is notably reduced. The process commences by the ODA process in our work we selected Haar Cascade since the required images are frontal face images only. The obtained results from the Haar Cascade yielded a 94% accuracy in face detection. The FaceNet has an accuracy of 99.63% therefore, the results of the face embedding extraction are highly accurate. Finally, the SVM predicts the identity of the user efficiently thus, the face recognition model reduced the EER results. As illustrated in Fig. 7A the EER is 0.22 which is the lowest obtained value in comparison to the related work as in Table 2.

Score level fusion consolidated the two biometric score matches to identify the user. It is commonly used since it always produces the best results in comparison to the other fusion techniques. The obtained EER in our work is 0.011 as illustrated in Fig. 7C which is the lowest scores in comparison to the related work as in Table 2. Despite, the voice recognition being the second to lowest, *Soltane (2015)* voice recognition yielded an EER of 0.0054 which is less than the obtained result in our work,it could potentially be due to the different frame sizing in voice samples. Nonetheless, our work has successfully surpassed the previous work in data fusion which is the most vital part in user authorization.

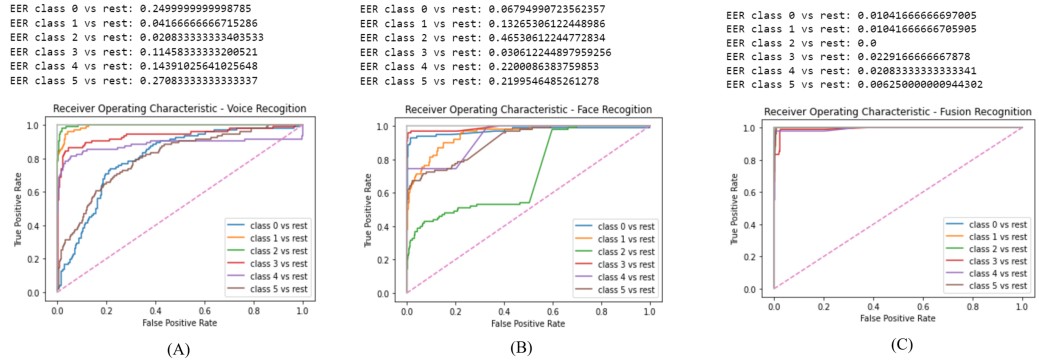

**Figure 7** ROC results: (A) ROC for voice recognition, (B) ROC for face recognition, (C) ROC for fusion recognition.

**Table 2** Comparison of the proposed scheme's obtained EER with other published results.

| Author | Database | Results (EER %) | | |
|---|---|---|---|---|
| | | **Face** | **Voice** | **Fusion** |
| *Chetty & Wagner (2008)* | AVOZES | 3.2 | 4.2 | 0.73 |
| *Palanivel & Yegnanarayana (2008)* | nEWSPAPERS | 2.9 | 9.2 | 0.45 |
| *Raghavendra, Rao & Kumar (2010)* | VidTIMIT | 2.1 | 2.7 | 1.2 |
| *Elmir, Elberrichi & Adjoudj (2014)* | VidTIMIT | 1.02 | 22.37 | 0.39 |
| *Soltane (2015)* | eNTERFACE | 0.399 | 0.0054 | 0.281 |
| *Kasban (2017)* | PROPOSED | 1.95 | 2.24 | 0.64 |
| *Abozaid et al. (2019)* | PROPOSED | 2.98 | 1.43 | 0.62 |
| Proposed scheme | AVSpeech | 0.22 | 0.13 | 0.011 |

From the results we observe that the proposed scheme indeed reduce the EER since, FaceNet automatically learns a mapping from facial image data to a condensed Euclidean space, where distances are directly mapped to a measure of facial similarity. Thus, having an accuracy of of 99.63%. However, the voice recognition utilizing GMM did not produce the lowest EER. The lowest fusion EER in our scheme mainly resides to the use of the FaceNet model which is the state-of-art model for face recognition.

## ACKNOWLEDGEMENTS

We like to thank (*Ephrat et al., 2018*) for providing the dataset for testing and training our system.

### Funding

The authors received no funding for this work.

## Competing Interests

The authors declare there are no competing interests.

## Author Contributions

- Bayan Alharbi conceived and designed the experiments, performed the experiments, analyzed the data, performed the computation work, prepared figures and/or tables, authored or reviewed drafts of the article, and approved the final draft.
- Hanan S. Alshanbari conceived and designed the experiments, performed the experiments, analyzed the data, performed the computation work, prepared figures and/or tables, authored or reviewed drafts of the article, and approved the final draft.

## Data Availability

The code is available at Zenodo:

bayansharbi. (2023). bayansharbi/Face-Voice-Biometric-authentication: v1.0.0 (v1.0.0). Zenodo. https://doi.org/10.5281/zenodo.7677169

The third-party data available is available at Looking to Listen at the Cocktail Party: A Speaker-Independent Audio-Visual Model for Speech Separation. https://looking-to-listen.github.io/. Contact: Ariel Ephrat.

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
