# Peer review of "Face-voice based multimodal biometric authentication system via FaceNet and GMM"

_PeerJ Computer Science, doi:10.7717/peerj-cs.1468_

## Round 0.1 · original submission · Major Revisions

Revise according to the comments of the reviewers.

Reviewer 1 ·

Basic reporting

I think the authors have summarized and motivated the problem really well. It is clear that they want to present a method for a multimodal biometric authentication system as using multiple modalities as input is better than a unimodal system.

They have referred to a lot of prior work in this domain too.

Experimental design

The method adopted by the authors has been described very clearly. However, I am not sure I understand how their method is different from the existing work?

Even in Table 2, they show that their method is performing better than 7 other related works. But, what exactly is causing this increase in performance.

So, it is important to analyze this performance with respect to their own method and other methods.

Validity of the findings

The dataset used is pretty big. Was the entire dataset used for producing these results? If not, then why?

Moreover, the dataset paper itself presented results on certain baselines using deep learning networks. I would believe using a deep learning network will perform much better on this task too. did the authors explore that?

Also, why have the results only been presented on 1 dataset? There are so many such audio-visual datasets that could also have been used for this task. This is important to report as this will speak to the generalizability of this proposed method.

Additional comments

- I think significant more experiments are required to show the validity of the work.
- Moreover an in-depth analysis on why this method is working better than the 7 other methods is required.

·

Basic reporting

This paper is well-written in English and well-organized. The authors proposed an improved multimodal authentication system using FaceNet and GMM to deal with face and voice signals. The authors conducted experiments on a public dataset and their proposed method achieved good performance on the dataset. But there are still some text descriptions that can be improved:
1. In sentence 68, can the authors add a figure of the unimodal biometric system? It will be helpful for readers to capture the difference between the unimodal and multimodal systems.
2. In sentence 85-86, authors wrote 'The rest of paper.....', but only section 2 is involved in the following texts. Other sections should also be involved here.
3. In sentences 159 and 161, 'Abozaid et al.' is duplicated.
4. For Figures 2-4, some descriptions of the workflow of the frameworks may need to be added.

Experimental design

Some questions for the experiments:
1. In sentences 192 196, the authors wrote "alpha = 0.98...." and "alpha_0 = 0.53836". Are there any experimental results that can support these parameter selections?
2. In Table2, the authors show some existing methods' results. Do all of the methods conduct experiments in the same dataset? If not, I think only the experimental results within the same dataset can be shown here.
3. For Figure 7 roc curve, auc scores may need to be added in the figure legend for all curves.

Validity of the findings

No comment.

---

## Round 0.2 · Major Revisions

Please make revisions to the paper based on the reviewer's comments.

·

Basic reporting

The improvement of the paper and the authors' responses have addressede my concerns.

Experimental design

The improvement of the paper and the authors' responses have addressede my concerns.

Validity of the findings

The improvement of the paper and the authors' responses have addressede my concerns.

Reviewer 3 ·

Basic reporting

The proposed scheme is presented in great detail, but the language needs substantial improvement.

Experimental design

The experimental design is logical and interesting.

Validity of the findings

The findings are neutral, and if possible, more state-of- the-art models should be introduced for comparison.

Additional comments

1) Language needs great improvement. A large number of typos are observed in the current manuscript. For example, in paragraph two, Section “Related work”, “Sarier (2021) proposed a a new multimodal biometric authentication” should be “Sarier (2021) proposed a new multimodal biometric authentication”. In paragraph one, Section “Methodology”, “In the process of authenticating he user voice and face samples” should be “In the process of authenticating the user voice and face samples”. In paragraph one, Section “Face Recognition”, “making the face-recognition process more challenging. .” should be “making the face-recognition process more challenging.”. In addition, there are many sentences are confusing. Please double check the whole manuscript.

2) More state-of- the-art models should be introduced for comparison. The methods currently compared by the authors lack the achievements of the last two years, so the reviewer is concerned about the superiority of the proposed method.

3) With biometric authorization at the core of this research, some novel work needs to be mentioned. If possible, an overview of class-imbalance privacy-preserving federated learning for decentralized fault diagnosis with biometric authentication is needed.

4) Many abbreviations are defined repeatedly. For example, in Section "Gaussian Mixture Model", MFCC and GMM have been previously defined.

5) The quality of Figs. 3 and 6 could be improved.

---

## Round 0.3 · accepted · Accept

According to the comments of reviewers, after comprehensive consideration, your submission can be published.

·

Basic reporting

I am satisfied with the modification of this paper.

Experimental design

I am satisfied with the modification of this paper.

Validity of the findings

I am satisfied with the modification of this paper.

Additional comments

The paper could be further enhanced by improving the descriptions of the formulas. For instance, in reference to Equation (7), the authors mention, "The results are too small". To provide a clearer mathematical context, it would be beneficial to define the value range of the variable. In this case, it could be stated as 'P(x|\theta) \in (0,1)'.

Reviewer 3 ·

Basic reporting

The proposed scheme is presented in great detail, and the language has been has been greatly improved.

Experimental design

The experimental design is logical and interesting.

Validity of the findings

The findings are neutral, and conclusions are also clearly stated.

Additional comments

All my comments have been well-addressed. This reviewer recommends acceptance of this manuscript.